# Peer review of "Recent Progress on Semiconductor-Interface Facing Clinical Biosensing"

_sensors, 2021, doi:10.3390/s21103467_

Round 1

Reviewer 1 Report

In the paper, the authors demonstrates that semiconductor based field-effect transistors are amazing enhancer gadgets due to their delicate interface towards surface adsorption. This results in application of these as a sensor and biosensors. Besides, the semiconductor material has enormous recognizable fixation extends, high affectability, high consistency for solid detecting and the ability to coordinate with other microfluidic gatherings.

The reviewer has few notes which should be commented or amended by the authors to better reveal the manuscript results.

1) The reviewer highly recommends to enhance the abstract by listing types of biosensors reviewed in the paper and principles of classification of these.  Besides, it is necessary to present results, obtained in the paper, in the abstract.

2) In order to better classify the biosensors, studied in the paper, it is necessary to analyze in more details the technical and metrological characteristics of state-of-the-art biosensors and further outlook of improvement of these. Comparison tables could be useful.

3) The reviewer suggests to make the conclusion in a more specific manner. The conclusion section should present new results obtained by authors as a result of the review.

4) A description and reference (mention) to a figure must be presented prior the figure itself. In the current version of the paper, a description nor reference is presented.

5) In the paper, the lines 588-614 and 513-540 present completely the same information.

6) In the paper, references are formatted in a different manner (e.g. line 162, reference [65] and line 160, reference 31).

7) For many chemical compounds indexes are not subscripts.

Reviewer 2 Report

In this manuscript, Wang and co-workers presented recent advance in the fabrication of semiconduction interface-based biosensor platforms for clinical biosensing. For this aim, they demonstrate important studies on the fabrication of interface-based FET, photoelectrochemical, optical, and metal-assisted SPR biosensors for biomedical determinations. In addition, the progress on the fabrication of organic interface-based biosensors for clinical applications was presented. It is a very interesting topic, which will be helpful for the fabrication of novel semiconductor-based nanodevices for biomedical applications. However, the organization of this manuscript should be improved. Therefore, I recommend a major revision for this manuscript.

Special comments for the revision:

  1. More information on the applications of semiconductor interfaces for other biosensors besides FETs should be added into the part of “Abstract”.
  2. Previously, a lot of reviews on the utilization of semiconductor materials for sensors and biosensors have been released, and therefore, it is necessary for the authors to add more discussion on the novelty and significance of this work by comparing with previous reports.
  3. The organization of the manuscript should be improved. For instance, in Part 2, the authors used the sub-section “Current progress”, which is not appropriate. It is suggested for the authors to present the sub-sections with the type of biosensors. Like this, “2. Semiconductor interface-based FET biosensors; 3. Semiconductor interface-based Photoelectrochemical biosensors; …”.
  4. The sub-title of part 3 is also not appropriate, which could be modified to “Organic semiconductor interface-based biosensors”.
  5. In Part 4, the authors should indicate clearly the potential and new research topics related to this field by providing their own viewpoints. The sub-title could be “Conclusions and perspectives”.
  6. The authors made a very comprehensive introduction of recent progress on the fabrication of various semiconductor interface-based biosensors, however, the suitable summaries are missing. Therefore, it is suggested for the authors to add a few tables to summarize the materials, fabrication techniques, sensor type, sensing performances, and others of the introduced biosensors.

Round 2

Reviewer 1 Report

In the revised paper the following issues are not completely eliminated.  

1) A description and reference (mention) to a figure must be presented prior the figure itself. 

2) In the paper, references are formatted in a different manner.

3) For some chemical compounds indexes are not subscripts.

Author Response

1) A description and reference (mention) to a figure must be presented prior the figure itself. 

we rearranged the figure meeting this comments mentioned

2) In the paper, references are formatted in a different manner.

we downloaded the MDPI style and managed the references

3) For some chemical compounds indexes are not subscripts.

we have gone through the paper and address each of the chemical names.

Reviewer 2 Report

In this revised verison, the authors made great improvement according to the comments and suggestions of the referee. I recommend the publication of this manuscript with the current version.

Author Response

Thanks for recommendation